# Fourier Synchrosqueezing Transform-ICA-EMD Framework Based EOG-Biometric Sustainable and Continuous Authentication via Voluntary Eye Blinking Activities

**DOI:** 10.3390/biomimetics8040378

**Published:** 2023-08-18

**Authors:** Kutlucan Gorur

**Affiliations:** Electrical and Electronics Engineering Department, Bandırma Onyedi Eylul University, 10250 Balıkesir, Turkey; kgorur@bandirma.edu.tr

**Keywords:** EOG-biometric, FSST, ICA, EMD, ensembled RNN-deep models, E-learning

## Abstract

In recent years, limited works on EOG (electrooculography)-based biometric authentication systems have been carried out with eye movements or eye blinking activities in the current literature. EOGs have permanent and unique traits that can separate one individual from another. In this work, we have investigated FSST (Fourier Synchrosqueezing Transform)-ICA (Independent Component Analysis)-EMD (Empirical Mode Decomposition) robust framework-based EOG-biometric authentication (*one-versus-others* verification) performances using ensembled RNN (Recurrent Neural Network) deep models voluntary eye blinkings movements. FSST is implemented to provide accurate and dense temporal-spatial properties of EOGs on the state-of-the-art time-frequency matrix. ICA is a powerful statistical tool to decompose multiple recording electrodes. Finally, EMD is deployed to isolate EOG signals from the EEGs collected from the scalp. As our best knowledge, this is the first research attempt to explore the success of the FSST-ICA-EMD framework on EOG-biometric authentication generated via voluntary eye blinking activities in the limited EOG-related biometric literature. According to the promising results, improved and high recognition accuracies (ACC/Accuracy: ≥99.99% and AUC/Area under the Curve: 0.99) have been achieved in addition to the high TAR (true acceptance rate) scores (≥98%) and low FAR (false acceptance rate) scores (≤3.33%) in seven individuals. On the other hand, authentication and monitoring for online users/students are becoming essential and important tasks due to the increase of the digital world (e-learning, e-banking, or e-government systems) and the COVID-19 pandemic. Especially in order to ensure reliable access, a highly scalable and affordable approach for authenticating the examinee without cheating or monitoring high-data-size video streaming is required in e-learning platforms and online education strategies. Hence, this work may present an approach that offers a sustainable, continuous, and reliable EOG-biometric authentication of digital applications, including e-learning platforms for users/students.

## 1. Introduction

Physiological and behavioral biometric traits can separate one individual from another and deal with identifying and verifying a unique person. These traits can be permanent, unique, and complex definitional in biometric authentication systems. The systems observe the security aspects of the applications that are imperative for security requirements of the digital world and technologies [1,2]. Biometric authentication systems have been steadily growing in popularity due to the development of many useful, challenging, and widely accepted applications, such as security issues and identity access management, such as cellphones, laptops, and entering a building [3,4]. These systems offer approaches to solve access control problems related to the authentication and verification stages [1,2,5].

Biometric authentication techniques implement behavioral or physiological features, including fingerprint, palm print, face pattern, iris, voice, and gait analysis to identify someone [6,7]. Physiological traits are known as the physical traits of an individual (fingerprint and hand etc.), and these static attributes are widely accepted because of their collectability and inexpensive characteristics in verification [1]. However, these static conventional biometric methods can be abused and sometimes can be easily accessed by imposters attacks [8,9]. Hence, security concerns against threats and fraud exist in these biometric approaches. On the other hand, in recent years, dynamic biological biometric traits are gaining popularity using EEG (electroencephalography), ECG (electrocardiogram), EMG (electromyogram), and PPG (photoplethysmography) signals [2,8,10]. All these 1D time-series signals have advantages and disadvantages in authentication systems. For example, lately, electroencephalography (EEG)-based authentication has received considerable attention from the scientific community. However, the limited usability of these biometric systems has been noted due to the high mental effort and continuous requirement of cognitive ability (hard-to-elicit motor imagery EEGs) [11]. ECG-based biometry will also succeed if fiducial points (P, QRS, and T complexes) are correctly identified in intrabeat and interbeat variations. However, correctly identifying fiducial points is reported as a difficult task in a practical biometric application scenario [8].

Electrooculography (EOG) is electrical activity resulting from a potential change due to eyeball or eyelid movements. The EOG waveform is recorded with skin electrodes around the eyes, or EEG electrodes are placed on the scalp. The amplitude of the EOG signals occurs between 10 and 200 µV and has been observed to change at frequencies falling within the 0.5–15 Hz range. The eyeball acts as a dipole and charged organ [12,13,14]. The positive and negative poles are oriented anteriorly (cornea) and posteriorly (retina). When the eyeball rotates about its axis, thus acting can generate a detectable amplitude electric signal that is distinguished by any electrodes near the eye or scalp electrodes. When the eyeball rotates upwards, the positive pole (cornea) closes to the frontal electrodes, producing positive deflection changes. Otherwise, when the eyeball rotates downwards (closer to the ear reference electrodes), it produces a negative deflection change. Deflections of eye blink resemble this situation. When the eyelid closes, the cornea closes to frontal electrodes, and a positive pulse occurs. However, the eyelid opens, and the cornea rotates away from the frontal electrodes, producing a negative pulse. The duration of eye blinking occurs in the range of 300 to 400 ms [14].

The proposed authentication process with FSST-ICA-EMD framework and voluntary eye blinking activity employs convenient methods (such as *one-versus-others* strategy and template matching) to confirm identity in the enrollment and authentication phases. In order to accomplish this, the following technical contributions were made:▪ This article proposes a *one-versus-others* biometric authentication approach investigating FSST-ICA-EMD framework using ensembled Recurrent Neural Network (RNN) models via voluntary eye blinking activities (EOG responses) to high correct recognition rates, reliable and suitable for next-generation consumer electronic devices. According to our best knowledge, this is the first attempt to explore high-level time-frequency features extracted by FSST and decomposed EOGs via ICA and EMD in a combined framework for EOG-biometric authentication in the existing literature.▪ Verifying the proposed authentication approach is essential via statistical discrimination to present a robust and effective system. Hence, this work is a first attempt to implement broad statistical methods to estimate discrimination EOG-biometric besides the correction rates (sensivibity, F-score, accuracy etc.) and TAR/FAR score metrics.▪ Visualizing the functional brain connectivity with circular graphs and brain mappings during eye-blinking activity. The following key superiority the FSST-ICA-EMD framework for EOG-biometric authentication focuses on the short-length of time (only 0.5 s) for training/testing/attempt processes for deep RNN models in enrollment and authentication. As far as we know, this time-segment is the least for the mentioned processes of the deep models. Thus, this advantage might address some of the aforementioned problems for swift enrollment and authentication phases [8]. The most used terms in the article are listed in Table 1.

The article is also organized as well: Section 2 discusses the literature review for this work. Section 3 addresses the methods and dataset descriptions. The results section is explained in Section 4. Section 5 reports the conclusion section to analyze outcomes and future work.

## 2. Literature Review

In recent years, the capability of eye-blinking EOG (electrooculogram) signals has been explored to discriminate between individuals in EOG-based biometric authentication systems. The eye-blinking EOG signals are extracted from brain waves recorded using an EEG headset. This new modality for authentication is a promising method. Furthermore, it has achieved a high recognition rate of up to >90.0% in identification mode and a low false acceptance rate (FAR) among multiple persons [12]. Moreover, a multi-level EEG-based authentication approach with EOG signals has also been reported in the limited literature [13,14]. Jalilifard et al. have developed a hybrid human authentication system using EEG and EOG signals generated by involuntary spontaneous blinking eye movements. Patterns of a series of blinks were processed by using Gated Recurrent Unit (GRU), and the classification accuracy was obtained over >98.0% [14]. In another study, Abo-Zahhad et al. provided a human recognition system voluntary eye-blinking waveform of around 97.0% in the test phase of identification mode. Linear Discriminant Analysis (LDA) was implemented to identify 25 individuals [15]. Juhola et al. has proposed a hybrid biometric system with EOG and VOG (video-oculography) signals using shallow machine learning models (kNN, LDA, etc.). This verification method employs saccadic eye movements to reliably distinguish a legitimate person from others in the range of 45-99% correct recognition rates [15]. Empirical Mode Decomposition (EMD) was applied to isolate EOGs from raw EEG signals using the decomposition process into Intrinsic Mode Functions (IMFs) [12,14].

In addition, eye blink features have also been implemented several other image and video-based research. Eye movement features (the motion, speed, energy, and frequency signal of eye blinks) were reported consisting of implicit and dynamic patterns that occurred on the microsecond temporal resolution of event densities [16]. Hence, these studies were offered to be used in eye-tracking applications [16]. The disadvantage of this kind of work, participants/students have to look at the monitor or camera consistently during the authentication process [17]. Especially this progress is a hard task for students in online e-learning/e-assessment education and it needs a high-speed internet [17]. In recent times after the COVID-19 pandemic, the need for sustainable and continuous online monitoring and authentication to e-learning platforms has evolved to seek accurate identification and verification for performing any behavior not authorized by the device on which students perform the activity [17,18,19,20,21,22,23,24,25,26,27,28,29]. Thanks to these important opportunities of biosignal-based authentication ways, e-learning platform courses/examinations may be provided as an effective verification in the learning-teaching experience [18]. Thus it might be widely accepted in all educational environments as a different alternative to traditional monitoring systems. In the limited literature on e-learning and authentication, most works concern monitoring technologies (webcams and microphones). However, according to our best knowledge, there is a lack of study implementation of EOG-biometric authentication technology as sustainable and continuous access for e-learning platforms. Digital Transformation (DT) must be adapted to competitive strategies and constantly renewed to succeed in the global competition of education and economics. The most significant of these innovations is secure access to the DT. Thus the strategic significance of different biometric authentication in today’s global education and a sustainable environment will be anticipated in growing cases day by day. Furthermore, short-time enrollment/authentication phases during secure and quick access to the digital e-learning platforms can also concentrate on the economic situation of “Sustainable Energy” [19].

This study aims to develop Fourier Synchrosqueezing Transform (FSST)-ICA-EMD framework-based EOG-biometric authentication approach using voluntary eye blinkings. The Fourier SST proposes a sharper time-frequency estimation by reconstruction of the coefficients for the time-frequency matrix [30]. The relevant time-series features are extracted by the Fourier SST (FSST), and this time-frequency representation can provide better performance for the detection of patterns in biosignals [31,32]. Independent Component Analysis (ICA) employes statistical technique background to decompose mixed multi-channel sources into statistically independent channels [33]. This technique decomposes each multi-channel recorded EEG/EOG signal set into temporally independent components [34]. Here, the ICA was the fundamental method to provide high true acceptance rate (TAR) and low false acceptance rate (FAR) metrics in multiple times attempts to the ensemble (Long Short Term Memory) models for EOG-biometric *one-versus-others* authentication. In addition, EMD was used to extract EOGs from the raw EEG signals to explore the accuracy and other performance metrics regarding the isolated eye-blinking waveforms [14].

## 3. Methods and Materials

### 3.1. Related Work and Motivation

This article proposes a robust and effective EOG-biometric authentication approach using voluntary eye blinking movements consisting of enrollment and authentication phases (see Figure 1). These waveform biometric traits protect private information stored in digital devices. The unique primary novelty of the processing of EOG signals is based on the FSST-ICA-EMD framework. Fourier SST can prompt high-resolution time-frequency feature extraction for time-series data [32]. On the other hand, ICA separates statistically independent components for multichannel recordings, and EMD is able to isolate EOG signals from EEGs [15,33]. The combined framework has highly capable of reliable and high-performance *one-versus-others* EOG-biometric authentication implementing ensembled RNN models. Majority voting, matching, and thresholding are also implemented to achieve consistent performance metrics. Time-segment generator extract random frames (0.5 s for 100 times) from the time-series data are to attempt these frames (for determining TAR and FAR scores) into the ensembled RNN models.

The applied statistical techniques in the work are:t-distributed stochastic neighbor embedding (t-SNE) was employed to demonstrate scattering for distinction.Probability density function (PDF) distribution for the distinction individuals (cross-subject and same-subject) over the diverse scalp parts.Correlation matrix for each subject to verify the unique pattern.Recurrence Plot (RP) is used for the seperation of sequential recorded channel source separation to verify the ICA/EMD technique effectiveness.MANOVA (Multivariate ANOVA) analysis was to describe grouping separation among individuals.Functional connectivity analysis was drawn to correlate each brain cortices over the Circular Graph technique.

### 3.2. Eye Blinking Activity and EOG Signal Acquisition

Eye blinking movement generated EOG responses (see Figure 2) were recorded in a noninvasive manner over the scalp with the Micromed SAM32RFO acquisition device. The impedance values were kept under the 10 kΩ. The sampling frequency was 1024 Hz and 50 Hz notch filter was applied to remove the power line noise [11]. After a filtering operation, the raw EOG signals are decomposed source separation via ICA and made extraction EOGs from brain activity via EMD according to the literature background (see Figure 1) [15,33]. The headset has 19 wet electrodes (see Table 2 for enumerated channels). The international 10–20 electrode placement system was employed with monopolar leads on the scalp (see Figure 3). Left-right earlobes (A1–A2) and left eyebrow are defined as the references and ground, respectively [11]. The indication of the brain cortex in the Figure 3 are titled as; F: *Frontal*, C: *Central*, T: *Temporal*, O: *Occipital* and P: *Parietal*.

### 3.3. Trial Organization

For this experiment, we settled on a single-trial organization consisting of multiple voluntary eye-blinking movements. The participants followed the experimental instructions on a monitor (17” LCD). These instructions are represented in Figure 4. Each experimental trial included different phases; starting point fixation, voluntary eye blinking movements, ready for relaxation, and resting state.

According to an experimental procedure, this trial organization begins with a start fixation point. Then the voluntary eye-blinking activities were carried out for 6 s without movement. After that, 1 s and 8 s periods are stated as the ready relaxation and resting state phases, respectively. The same process is repeated six times, and the entire trial is conducted, lasting 86 s duration. Therefore, the dataset size for eye blinking acting based EOGs is recorded as 36,864 × 19 (6 s × 1024 sampling rate × 6 times × 19 channels) for each participant during a single trial. In the resting state phases, the participants were instructed to simultaneously think of hand-action motor imagery (EEGs). However, the resting state phases were cut out to define EOG activity solely in the developed EOG-biometric authentication.

Naive and healthy subjects (aged from 20 to 38) were recruited for this experiment, including students and academic staff at the Bandirma Onyedi Eylül University. One focus point is to explore the FSST-ICA-EMD framework with limited users in EOG-based authentication. Furthermore, male (3) and female (4) subject numbers were chosen in close proportions to prevent from gender bias. Reliably distinguishing legitimate student children with electrooculography is suitable in computational verification due to the easy-to-use and universal characteristic of eye blinking activities.

### 3.4. Time-Frequency Feature Extractıon and Ensembled RNN Models

Feature extraction is reported as a critical step in developing effective and robust biometric systems. This Fourier Synchrosqueezing Transform (FSST)-based feature extraction strategy was proposed to extract time-frequency domain features for each time segment (0.5 s) of eye-blinking EOG signals. Then time-frequency matrices (based on FSST) were employed to enable ensembled RNN models for the one-versus-others biometric authentication systems. FSST reorganizes energy only in the frequency direction and is a highly effective way. Therefore, the time resolution of EOG signals can be effectively maintained in the time-frequency domain. As far as we know, in terms of the available literature background, this FSST-based feature extraction technique is used to improve EOG-biometric authentication performance for the first time in this research study. The time-frequency feature extraction process is shown in Figure 5.

Ensemble methods are a technique that combines predictions of various machine learning models, which helps classification models to have stronger generalization ability. Thus, better prediction performance has been achieved in comparison with the base classifiers. Reducing the model’s bias and the dependence of the predictions on the properties of a training set is important and is reduced by a few coupling models.

In our study, the ensembled model approach was implemented for each person (*one-versus-others*) in distinctive form datasets (one feature set-versus-others’ feature sets). These trained machine learning models were fed with complementary sets in terms of the data size that can produce diverse and complying outcomes (own personal/the one or others/the others) [35]. Adversarial-trained LSTM models (LSTM model-1/LSTM model-3) (see Figure 6) are working against each other. And the rest one (BiLSTM model-2) was fed with equal-size features. Finally, the last prediction is conducted by a majority voting process [36,37]. The testing phase was carried out using feature set-2 on the ensembled RNN models.

Time segments and training size are important critical parameters to provide better biosignal-based biometric systems [38]. The FSST conducted the feature extraction process (see Figure 5). The relevant time segment is defined as 0.5 s over the single-trial EOG sequential time series for the training, testing, and attempts on the ensembled RNN models. Biosignal-based biometrics does not describe the optimal time segment length. Nevertheless, increased time segments are noted that this change only achieves better performances significantly. Furthermore, increased training size can achieve higher biometric system performances, as in the reported works [39]. In our study, an 80–20% rate of data splitting strategy was employed for the training and testing phases, respectively. According to the literature, this rate is the best and optimal rate for providing higher performance for biosignal-biometric systems constructed on RNN deep models [8].

### 3.5. Fourier Synchrosqueezing Transform (FSST)

The analysis of non-stationary electrophysiological signals requires high-level time-frequency (TF) methods due to the spectral properties of these time-series signals varying over time [30]. Synchrosqueezing transform is an intended advanced TF strategy based on Continuous Wavelet Transform (CWT) and the Short-Time Fourier Transform (STFT) to concentrate sharpens localization of spectral estimation for the oscillation components of a signal. The synchrosqueezing process reassigns “condenses” time-frequency maps around the curves of the instantaneous frequency changes [30]. Hence, FSST is capable of concentrating energy coefficients to obtain compact TF distribution covering TF-based feature extraction vectors. Thus, it may enable high accuracy rates for EOG-based biometric authentication tasks (see Figure 7). STFT determines Fourier transforms while moving along the signal from start to end with a fixed-length window function [30,31]. The related mathematical relations were represented in the following equations. Where R and Re are defined for the round and real part operations, respectively.
(1)$$S\left(m,k\right)=\overset{N-1}{\underset{n=0}{\sum }}x\left(n\right)W\left(n-m\right)e^{-j\frac{2\pi }{N}k\left(n-m\right)}$$
(2)$$\;\hat{w}\left[m,k\right]=\left\{\begin{array}{c} R\left[Re\left(\frac{N}{2\pi j}\frac{S\left(m+1,k\right)}{S\left(m,k\right)}\right)\right]\;\;\;\;if\;S\left(m,k\right)\neq 0\\ 0\;\;\;\;\;\;\;\;\;\;\;\;\;\;\;\;\;\;\;\;\;\;\;\;\;\;\;\;\;\;\;\;\;\;\;\;\;\;if\;S\left(m,k\right)=0 \end{array}\right.\;\;\;$$
(3)$$\;FSST\left(m,\hat{k}\right)=\overset{N-1}{\underset{k=0}{\sum }}S\left(m,k\right)\delta \left[\hat{k}-\hat{w}\left(m,k\right)\right]\;\;$$

### 3.6. Independent Component Analysis (ICA)

Characterizing EOG time-series responses over the multisite EEG recordings is important for developing more reliable and better performance biometric authentication strategies during signal procedures in a framework [40,41]. Multiple electrodes over the scalp are composed of a multiple mixture of independent sources/electrical signals, which occur from the cerebral and/or extracerebral sources [42]. On this point, Independent Component Analysis (ICA) signal processing technique is so effective way to decompose it into a limited number of independent components (ICs) [43]. Here, it defines the multiple channel signals, while the coefficients matrix/mixing matrix and s describe the source components. Firstly, matrices are calculated with maximum likelihood estimation, and ICs are determined by multiplying the inverse of A by, as seen in the following equations:(4)X=As
(5)$$s=A^{-1}X$$

ICA has an important potential application to be considered as a powerful statistical technique for multichannel responses of a unique pattern while decomposing mixed signals into statistical ICs [34].

Recurrence plot (RP) is a recent analysis technique for nonlinear and dynamic activity of biological signals. This method can focus on the repeating patterns of the time-series signals over states and can define geometric structures to visualize the recurrent occurrences of states as a phase space [44]. In this research, RP was employed to visualize the decomposition success over the sequential channels (Fp1–Fp2) for the raw EOG signals and decomposed responses via ICA and EMD (see Figure 8). Long vertical line is able to describe responses, occurred the same state during some time steps [45]. Here, the related segments of this case for signals mean staying in the same phase space region for a short while, and it can be dominated by slow (theta or alpha) waves, including EOGs (8–15 Hz). Thus, high-frequency components can vanish, and slow waves are shown as longer times leading to the up-slope/down-slope patterns [45].

The Figure 8 represents the raw data signals and ICA/EMD applied responses, respectively/top-to-bottom. Moreover, the relevant signals are plotted from the Subject-3 during 1 s and distinct responses are to show the success of ICA/EMD decomposition techniques over the sequential channels.

### 3.7. Empirical Mode Decomposition (EMD)

The Empirical Mode Decomposition algorithm is considered an adaptive and data-dependent technique to decompose non-linear and non-stationary signals into a set of amplitude-modulated and frequency-modulated components called Intrinsic Mode Functions (IMFs) [14]. This separating analysis way is a powerful time–frequency analysis technique expressing the internal modes’ instantaneous frequency. IMFs include relevant and instant frequency information to obtain frequency values that change over time to offer better time resolution compared to ones. Thus, EMD is to facilitate EOG isolation from EEG signals/ and it is employed voluntary eye blinks extraction from the scalp-recorded EEGs in EOG-based biometric authentication [14,41].
(6)$$\;\;x\left(t\right)=\overset{k}{\underset{m=0}{\sum }}IMFm\;\left(t\right)+rk\left(t\right)$$

In the relevant formula, *k* defines the IMF number, and *rk (t)* stands for the final residual value. In this work, the eye blinking waveforms were extracted using EMD according to following last four IMFs and residual value (Isolated EOGs = IMF-2 + IMF-3 + IMF-4 + IMF-5 + Residual) (see Figure 9) [14].

The plotted signals in Figure 9 are recorded over the Fp1 channel and Fp2 channel in the left and right side, respectively. *X*-axis is to show the time in seconds and the *Y*–axis is the amplitude in µV.

### 3.8. Recurrent Neural Network (RNN) Deep Models

RNN-based deep model networks are able to perceive patterns from the sequential data via the learning of previous experiences when they encounter over time [46]. This ability provides a better recognition level for biological-based time-series signals, including EEGs. The existence of a correlation between the sequential time-series signals stored at different time intervals is named the term ‘long-term dependence’ [47]. Storing and transmitting the state information aims to process the sequences in the RNN deep models. LSTM is designed as a special type of RNN deep model and is capable of observing a correlation between the data learned [48].

On this point, if xi∈ℝd are defined as a time-series input vector x1,x2,…,xk and Long-Short Term Memory (LSTM) is able to generate a hidden sequence h1,h2,…,hk for each processing step [49,50]. The current input function (xt) and the previous hidden state/feedback into the neuron (ht−1) are both formulating the activation of the hidden state at time *t* [50]. Hence, this model is different from the CNN (Convolutional Neural Network) deep model [51]. 2D-CNN deep models extract deep features from the images [52] and are evaluated in image processing applications [53]. This related process is as the following term:(7)ht=fxt,ht−1

LSTM deep model has three modules in terms of the different gate combinations. The names of these gates are the forgotten $f_{t\;}$, input $i_{t}$ and output gates $o_{t}$. The transmitted size of information is determined for the next stage in the forgotten gates implementing the sigmoid function in this operation. The next step task is to decide the relevant information to be stored with the sigmoid function at the input gate. Then, the new state information is carried out for the memory cell in the system output [47]. The mathematical formulas are as follows:(8)$$i_{t}=\sigma \left(W_{i}\cdot \left[h_{t-1},x_{t}\right]+b_{i}\right)\;$$
(9)$$\;{\tilde{C}}_{t}=\tanh\left(\;W_{C}\cdot \left[h_{t-1},x_{t}\right]+b_{C}\right)$$
(10)$$C_{t}=f_{t}*C_{t-1}+i_{t}*{\tilde{C}}_{t}\;$$
(11)$$o_{t}=\sigma \left(W_{o}\cdot \left[h_{t-1},x_{t}\right]+b{\;}_{o}\right)$$
(12)$$\;h_{t}=o_{t}*\tanh\left(\;C_{t}\right)$$
where, according to the general terms, W is the weight vector, b defines the bias term, σ stands for the sigmoid activation function to put non-linearity. Hence, the new status information of the memory cell is calculated. Moreover, Bidirectional-LSTM (Bi-LSTM) calculates the sequential data in both directions (forward and backward propagation). In the first step, the model performs from the first value of the sequence to the last value. Then, the starting point is performed from the last time value to the first time value (see Figure 10) [47]. The models have been designed with multiple neuron numbers (200) and a fully connected layer (1 neuron). Finally, a softmax classification layer was added to the architecture. The activation function was ReLU, and the optimizer was chosen as Adam. The initial learning rate was defined as 1 × 10^−3^. The training process epoch was selected as 300. Table 3 concerns a literature survey summarizing EEG-based authentication and limited EOG-based authentication research in recent years, including RNN-based deep models.

### 3.9. Performance Metrics

The performance metrics are reported as a key to evaluating machine learning models in pattern recognition tasks and finding the optimal method for solving a problem. The selection of appropriate metrics is critical in data science [48]. A confusion matrix (see Figure 11) is employed to create the other metrics frequently from the fundamental metrics (true positive/TP, true negative/TN, false positive/FP, false negative/FN).

The number of correct predictions to the total number of predictions yielding the quality metric called classification accuracy (ACC). However, ACC is not accepted as an adequate parameter to determine the proposed algorithm’s performance totally. Hence, the other statistical performance metrics are described, including specificity (SPEC), sensitivity (SENS), precision (PREC), and F-score in this respect [48]. The F-score is the coherent coefficient of the precision and sensitivity [59]. The notations are below:(13)$$ACC=\frac{TP+TN}{TP+TN+FP+FN}$$
(14)$$SPEC=\frac{TN}{TN+FP}$$
(15)$$SENS=\frac{TP}{TP+FN}$$
(16)$$PREC=\frac{TP}{TP+FP}$$
(17)$$F-Score=2X\frac{PREC\;X\;SENS}{PREC+SENS}$$

Another important metric is the Information Transfer Rate (ITR) to express the amount of information transferred per unit time, especially for Brain-Computer Interfaces area. The mathematical formula is based on the logarithmic:(18)$$B=log_{2}N+Plog_{2}P+\left(1-P\right)log_{2}\frac{\left(1-P\right)}{\left(N-1\right)}$$
where B stands for the number of bits per trial, *N* is indicating class number, and the correctness of prediction is names as P [47]. The ROC curve is plotted to visualize graph of the false-positive rate (*X*-axis) and a true-positive rate (*Y*-axis) and The area under the ROC curve provides the AUC value between the interval [0, 1]. The greater AUC value is regarded as better discrimination [47].

In order to calculate the robustness of the biometric authentication systems, the ‘True Acceptance Rate’ (TAR) and ‘False Acceptance Rate’ (FAR) are implemented in the developed biometric approaches. Observation of the quality of specific attempts (genuine or forgery) is carried out to take into account verifying a true identity claim (TAR) and the percentage of unauthorized users (FAR) [40]. The expected and successful biometric system is designed to maximize the TAR score while minimizing the FAR score during multiple times of attempts. In this EOG-based biometric approach, 100 attempts were conducted to obtain TAR and FAR scores. Thus, the decision for calculating the TAR and FAR scores was provided after the matching and thresholding process according to the relevant formulas:(19)$$Decision\;of\;TAR\;\left(X\backslash Sm\right)=\left\{\begin{array}{c} Genuine,\;\;\;\;\;\;\;\;if\;\;Sm>th\\ Imposter,\;\;\;\;\;\;\;\;\;otherwise \end{array}\right.$$
(20)$$Decision\;of\;FAR\;\left(X\backslash Sm\right)=\left\{\begin{array}{c} Imposter,\;\;\;\;\;\;\;\;if\;\;Sm>th\\ Genuine,\;\;\;\;\;\;\;\;\;\;\;\;\;otherwise \end{array}\right.$$
where *X* and *th* are the query user, and threshold value, respectively. When the similarity matching (*Sm*) for TAR score is higher than defined *th* value, the user is treated as genuine else rejected. Otherwise, if the similarity matching (*Sm*) for FAR score is obtained higher than the defined *th*, the user is treated as an imposter or else rejected [40]. Threshold levels according to the similarity matching were described as 90% (TAR score) and 80% (FAR score), respectively.

## 4. Results

This work aims to analyze the performances of seven subjects for the FSST-ICA-EMD framework-based EOG-biometric authentication approach. In order to determine the robustness and outcomes of the *one-versus-others* biometric authentication system, ACC, SENS, SPEC, PREC, F-score, ITR, and AUC scores were determined, as well as finding the TAR and FAR metrics using the ensembled RNN-deep models.

Table 4 highlights the prediction performances of seven participants in the proposed *one-versus-others* EOG-biometric authentication scheme for isolated EOG signals generated by voluntary eye blinkings. According to the results, Subject-7 has achieved high classification performances for all metrics (ACC: 99.99%, SENS: 99.95%, SPEC: 100%, PREC: 99.99%, F-score: 99.95%, AUC: 0.999 and ITR: 0.931) among the others. The worst biometric classification discrimination was calculated on Subject-5 (ACC > 89%, AUC > 0.7, and ITR > 0.9 score). The framework attained the second highest classification performances provided over the Subject-4 (ACC: 99.93% and AUC: 0.997), Subject-1 (ACC: 99.92% and AUC: 0.997), and Subject-3 (ACC: 99.91% and AUC: 0.996), respectively. These biometric prediction results have depicted the number of four individuals reaching up to the >99% (ACC, SENS, SPEC, PREC, and F-score) discrimination regarding the observation metrics. Moreover, the average of these persons’ outcomes has been obtained over >96% accuracy, >0.9 AUC score, and >0.9 ITR value.

Because of the demand for secure user authentication systems, the stable EOG-biometric authentication approach to one specific individual is required under the diverse frequency and speed of eye blinks. In the limited existing literature for EOG-biometric systems, it is reported that spontaneous eye blinkings under a normal physical and emotional state can be influenced by diverse emotions, fatigue, and level of focus as well as the age or light condition. Table 5 defines the proposed system’s robustness against its own attempts and imposter attacks in verification mode using two metrics (TAR and FAR). According to the outcomes, the highest scores for TAR were yielded with Subject-3 (100%) and Subject-4 (100%). The other best discrimination (99 attempts out of 100) in the *one-versus-others* biometric authentication for the own attempts are observed in Subject-2, Subject-5, and Subject-6. According to the multiple time sequences of voluntary eye blinks and discrimination performances, Subject-1 and Subject-7 have achieved the least TAR scores at 98% and 96%, respectively. Furthermore, the following FAR metric scores show that Subject-1, Subject-4, and Subject-7 have 0 average values out of 100 attacks for each isolated EOG-based authentication system. The other subjects’ own systems have 3.33% (Subject-2 and Subject-5), 1.83% (Subject-3), and 0.5% (Subject-6) in terms of the FAR scores.

The most important technique for EOG + EEG signal decomposition for multiple channels is Independent Component Analysis (ICA). Table 6 presents the prediction results of the EOG-based biometric authentication with EEGs implemented ICA for multiple channel recordings. By analyzing the decomposed multiple channels for EOG + EEG responses in the *one-versus-others* authentication, it defines that all individuals have achieved >99% discrimination accuracies for each authentication case. Totally, the average accuracy, sensitivity, specificity, precision, and F-score prediction outcomes are also higher than 99% value. AUC and ITR scores are also greater than 0.99 values. However, the best discrimination is observed in Subject-4 in terms of all performance metrics (ACC: 99.93% and AUC: 0.997) during vertical voluntary eye blinkings.

For the proposed high-level feature engineering with FSST in the EOG-biometric authentication approach, the calculation of the matching score was revealed in Table 7. According to the TAR and FAR scores, the attempts and attacks for the own individual biometric authentication system, including ensembled RNN models, have shown high TAR scores (>99.57%) and low FAR scores (0%) on average. The least matching score for TAR score was obtained in Subject-6 (97%), and the others have provided 100% authentication success out of 100 attempts.

Following the steps above, the behavioral biometric feature of EOG signals recorded over the EEGs was revealed to recognize bioelectrical signal patterns during voluntary eye blinks for person authentication. The following tests compare EOG and EOG-EEG signals in authentication achievement consisting of the discrimination metrics and matching scores (see Figure 12 andFigure 13). Hence, it is clear to be in the observation that EOG-EEG signals have relatively better authentication success, especially in TAR (0.86% higher) and FAR (3.33% lower) similarity scores due to the multi-modality responses. Uniqueness and the biometric characteristics of each subject may be defined better by these two physiological traits.

The t-distributed stochastic neighbor embedding (t-SNE) was employed to specify the discrimination of each individual on the scattering benchmark for comparison [40] [60]. A unique colored cluster represented each subject. It also confirmed that the seven individuals had great separable outcomes based on voluntary eye blinkings. Each subject is represented by a unique colored cluster [61,62]. It was clearly observed that there are high intraclass similarity and low interclass similarity with minimal overlap for diverse subjects in the ICA-based dataset, ICA-EMD-based dataset, and FSST-ICA-EMD-based datasets (see Figure 14). In addition to seven-person separation via t-SNE, the discriminative ability of the *one-versus-others* biometric features were represented in Figure 15.

The probability distribution functions (PDFs) of the correlation coefficients between the frontal-lobe electrode locations (Fp1–Fp2 and Fp1–F3) aim to indicate the ability to discriminate among paired subjects. The largest differentiation (the smallest overlap area) means better authentication discrimination, and the smallest differentiation (the largest overlap area) has poor discrimination performances [40]. According to the PDFs, the lowest discrimination among the cross-subjects and same-subjects was obtained between Subject-1 and Subject-2. The highest separability performances were provided with Subject-6 and Subject-7 (see Figure 16). Correlation coefficient matrices might offer to provide a unique pattern for each subject and the relationship between affective discriminability-related cerebral cortex (see Figure 17). These results demonstrate the feasibility and potential robustness of this framework against to intra-session and inter-session variability. Furthermore, the least electrode pairs of related cortices may lead to unique information for each person and the usability of this FSST-ICA-EMD framework-based EOG-biometric authentication approach [40].

Multivariate analysis of variance (MANOVA/ multivariate extension of ANOVA) was implemented to compute group significance and to distribute each person among various FSST time-frequency feature groups [63]. The discrimination ability of the FSTT-based framework helps to distinguish each individual by these features. Clustering subjects via MANOVA shows the high group significance separation over the defined scores for each group (Figure 18) [39]. Hence, *p*-values were determined as less than or equal to 0.05, and these values offer to represent a significant difference among the individuals in terms of the feature-extracted EOGs generated by voluntary eye blinks. The distinctive ability of the features/framework is to maximize for each person, and the MANOVA scores were considered statistically relevant for further analysis in this *one-versus-others* EOG-based biometric authentication. The length of the EOG segments for training/testing phases in the ensembled RNN-based deep models might be investigated to provide less amount of data during enrollment and authentication progress. Thus, a biometric system can be user-friendly due to the minimized evaluation time. MANOVA can be employed to observe the influence of the performance of the EOG-biometric system over the significant scores in different segment lengths [8,39].

The time-series EOGs responses occur in frequency falling in the range of 0.5–15 Hz/20 Hz and have dynamic and nonstatic characteristics acquired by the origin of the voluntary eye blinkings [14,64]. The channel positions on the scalp were estimated to explore the performances of the raw EOG signals in the FSST-ICA-EMD framework-based biometric authentication. A minimal subset of recorded channels is targeted for low computational cost and portability of the equipment in biometric systems [3,40]. These responses correspond to the seven subjects and represent the brain mappings of the unique pattern for each person. The power spectral density outcomes for 15 Hz were plotted to address the temporal-spatial patterns. According to these mappings (see Figure 19 [65]) and functional connectivity (see Figure 20), frontal (high level) and posterior parietal (low level) cortices reveal the action-specific sites during eye blinking activations.

Furthermore, Circular Graphs (CGs) were described to calculate the correlation of scalp locations introducing the dynamics of functional connectivity across multiple channels [66,67]. Visualization functionality during voluntary eye blinkings for the biometric systems is important to observe nonlinear connectivity in every EOG recording. The connectivity values between channels were handled to draw the lines connecting the two channels. The drawn width on the circumference of the circle is to represent the correlation strength of the connection [68]. The lines were highly connected, clearly marking the frontal-to-other cortices in both hemispheres.

The major drawback for EEG-biometric systems is the instability issue over time due to the environment and mental state fatigue [40]. However, voluntarily generated EOG responses are much more stable than EEGs, leading to more reliable authentication progress. The other primary advantage of this FSST-ICA-EMD framework-based EOG-biometric authentication is to offer fast enrollment and authentication time with high TAR (≥98%) and low FAR (≤3.33%) scores via processing low time-segment (0.5 s). In the limited existing biosignal-related biometric research, few of them concern low-time segment conditions [40,56].

## 5. Conclusions

To address some of the aforementioned issues, we have developed robust (high TAR scores: ≥98% and low FAR scores: ≤3.33%) and high verification accuracy (ACC: 99.99% and AUC: 0.99) FSST-ICA-EMD framework-based EOG-biometric authentication approach using ensembled RNN-deep models during voluntary eye blinking activities. Our objective was to form as voluntary straightforward verification task as possible since this robust framework-based EOG-biometric approach has enabled a 0.5-s short measurement (time-segment) in training/testing/attempts of a single trial. Thus, it may be capable of high access enrollment/authentication phases in real-time usage in future research. With the COVID-19 pandemic and quarantines, the education system has been abruptly transformed into distance and online courses under e-learning applications [17,18]. Hence, FSST-ICA-EMD framework-based EOG-biometric authentication to this e-learning platform may offer sustainable, continuous, effective strategies and procedures.

### Future Work

Different adaptations and experiments have been left for future work due to a lack of time. Firstly, the limitation of this work has multiple electrode usage. However, this challenge can be updated in future works by considering the functional connectivity plots and brain mappings for limited and effective channels with dry and Bluetooth electrode technologies. In this biometric system, RNN-based ensembled models were implemented for better-recognizing time-series EOG signals. Moreover, leveraging transfer learning can be applied to state-of-the-art deep learning models and other biosignals/images to provide multimode biometric strategies [69,70]. Also, pre-trained networks and fuzzy-based expert system can be investigated on this EOG-biometric system [71,72]. Furthermore, real-time authentication experiments can be done on the students while accessing the e-learning platforms.

## Figures and Tables

**Figure 1 biomimetics-08-00378-f001:**
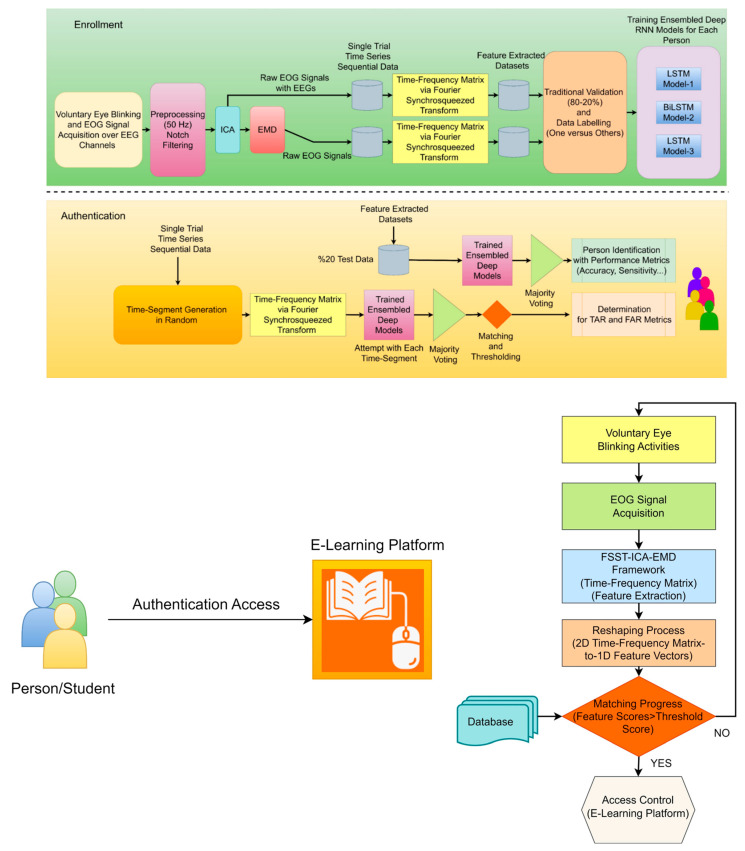
The scheme of FSST-ICA-EMD framework-based EOG-biometric authentication over voluntary eye blinkings/vertical saccades (**upper**), an approach for e-learning platform access and flow chart algorithm (**lower**).

**Figure 2 biomimetics-08-00378-f002:**
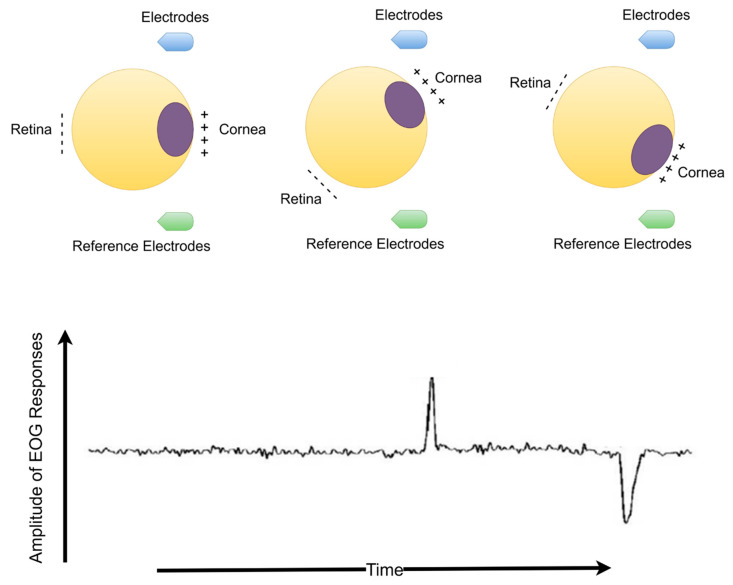
The dipole model of eyeball with EOG waveforms (Eye blinking waveforms during eyelid opening and closure have similar activity responses in terms of the generation).

**Figure 3 biomimetics-08-00378-f003:**
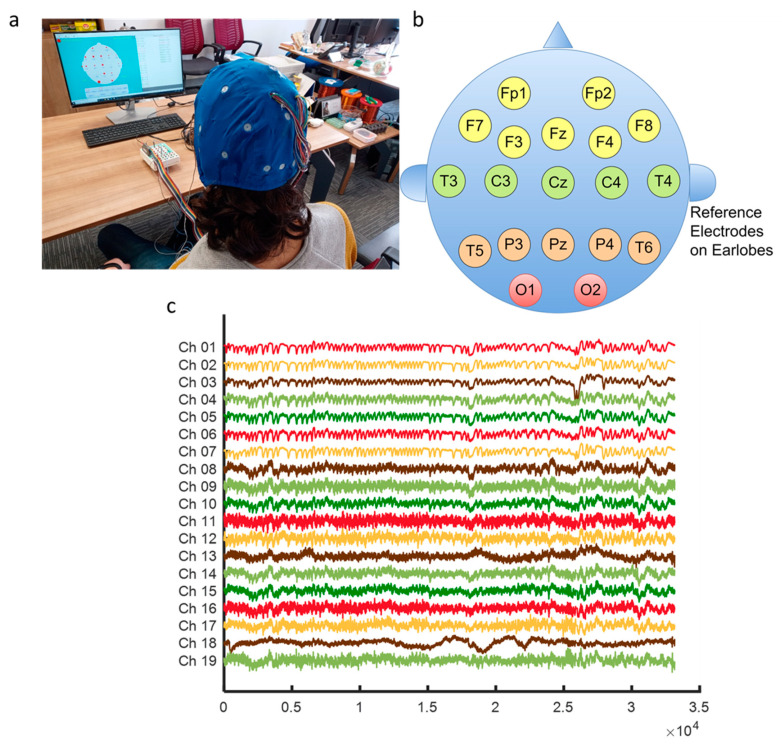
Signal acquiring scheme for EOG-biometric authentication with voluntary eye blinkings (**a**) Electrode placement setup on the scalp (**b**) Indication of brain lobe locations (**c**) Recorded EOG responses (from the Subject-5 samples).

**Figure 4 biomimetics-08-00378-f004:**
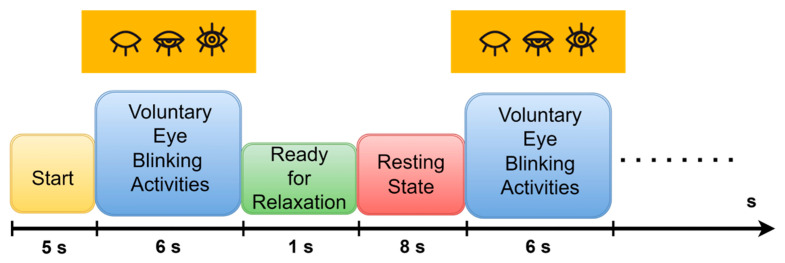
Experimental procedure for recording voluntary eye blinking activities in single-trial phases.

**Figure 5 biomimetics-08-00378-f005:**
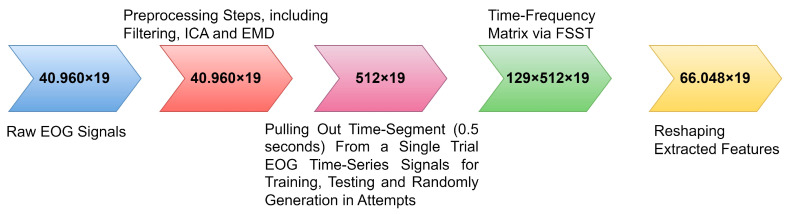
Spectral feature engineering technique via FSST for temporal-spatial pattern.

**Figure 6 biomimetics-08-00378-f006:**
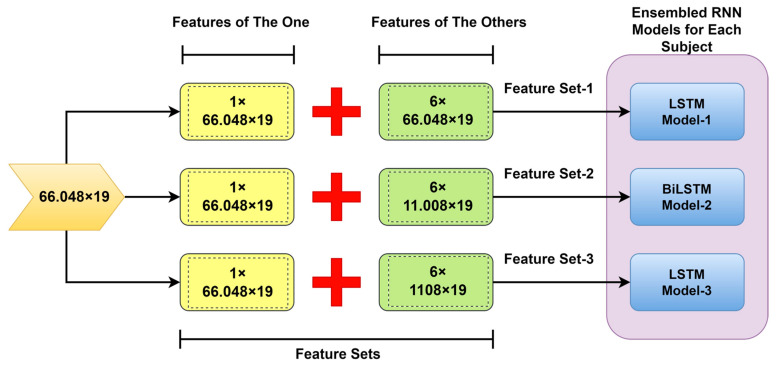
Concatenating the diverse features in the proposed one-versus-others authentication.

**Figure 7 biomimetics-08-00378-f007:**
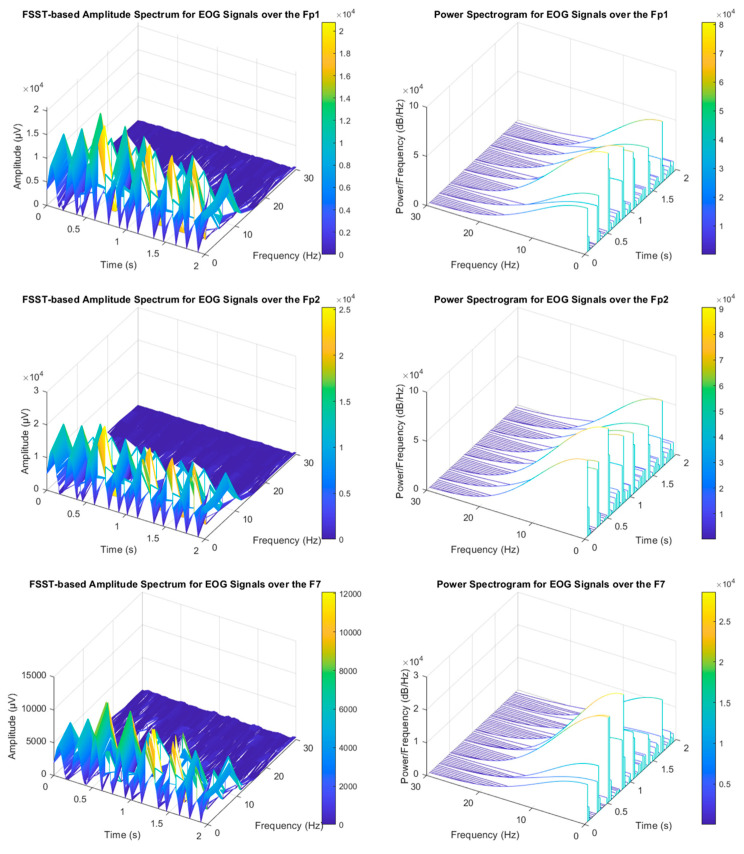
Voluntary eye blinking activities (The relevant EOG signals were collected from the Subject-3).

**Figure 8 biomimetics-08-00378-f008:**
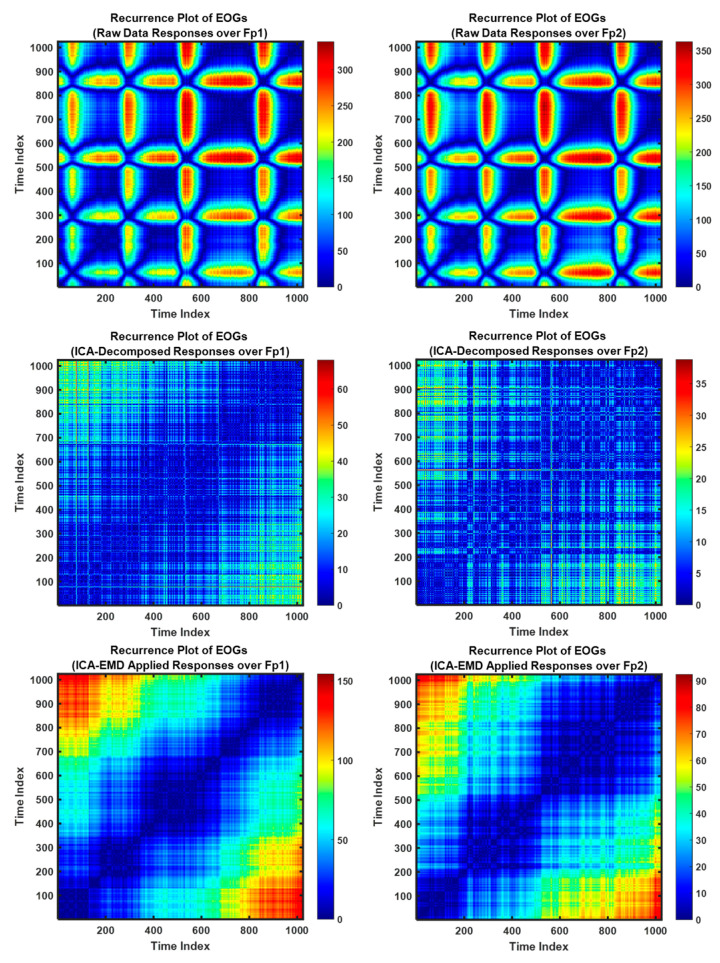
Recurrence plots of EOGs generated via voluntary eye blinkings.

**Figure 9 biomimetics-08-00378-f009:**
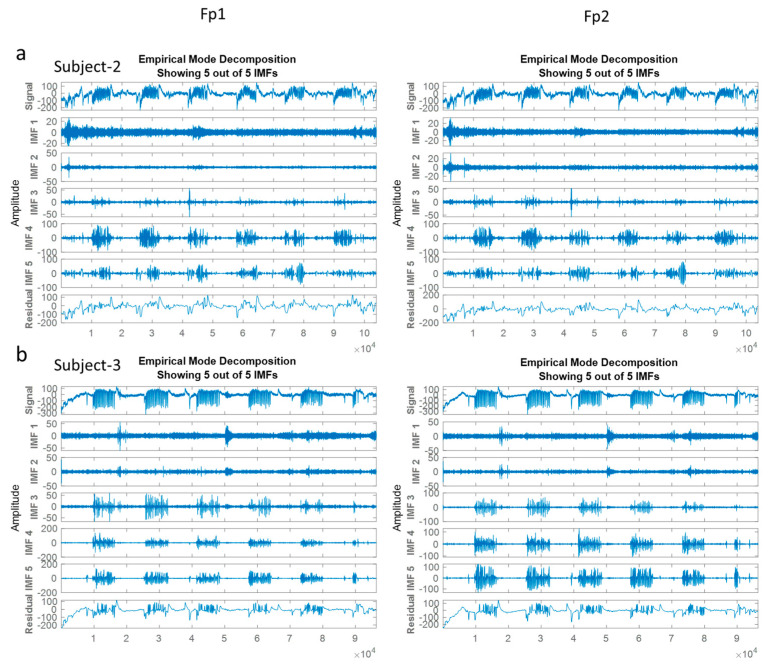
Isolation process of EOG components from the EEGs via EMD.

**Figure 10 biomimetics-08-00378-f010:**
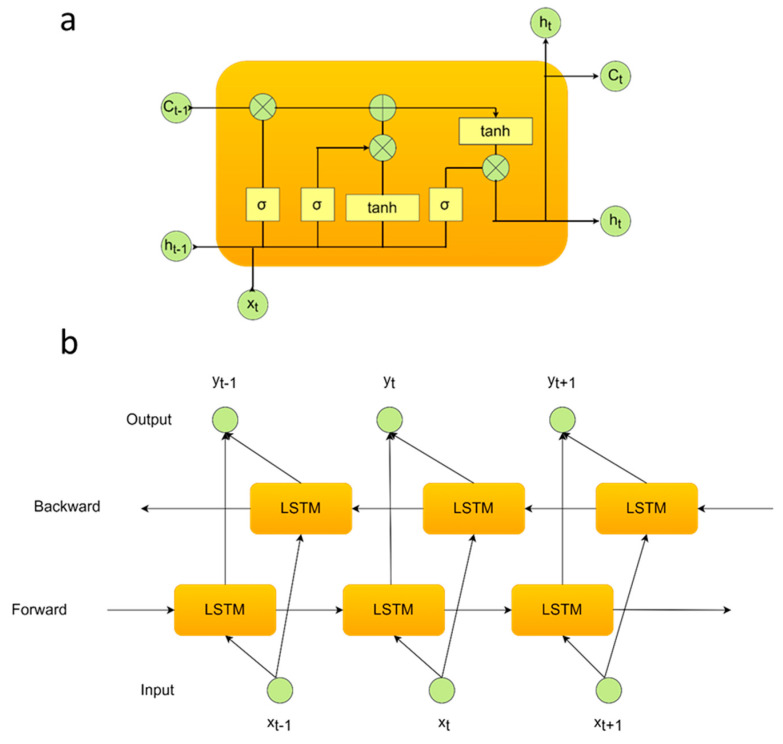
The internal architectures of the (**a**) LSTM and (**b**) BiLSTM deep models.

**Figure 11 biomimetics-08-00378-f011:**
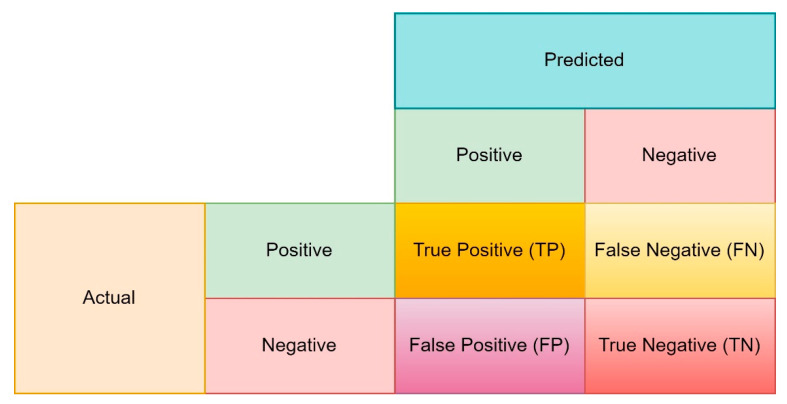
The confusion matrix schematic.

**Figure 12 biomimetics-08-00378-f012:**
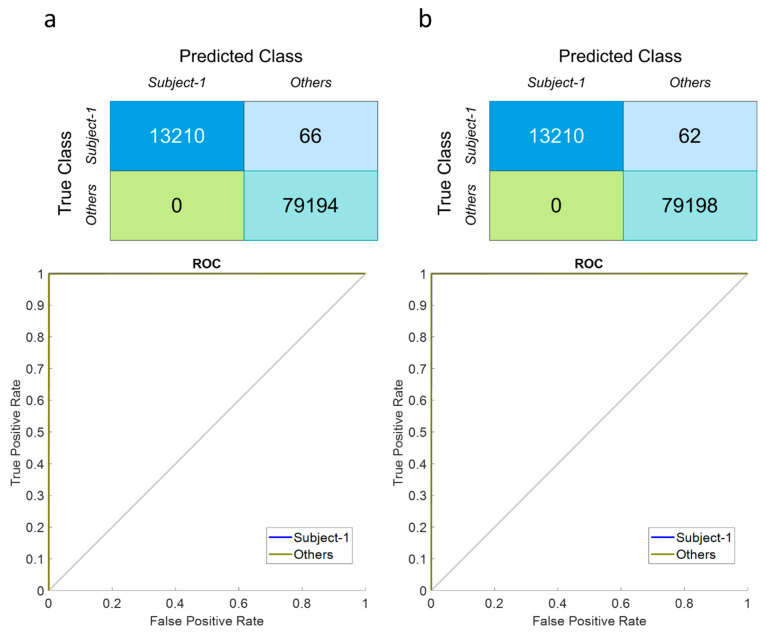
Confusion matrices and ROC curves for EOG-based authentication approach generated via voluntary eye blinks (**a**) Isolated EOG responses (**b**) Raw EOG responses with EEGs.

**Figure 13 biomimetics-08-00378-f013:**
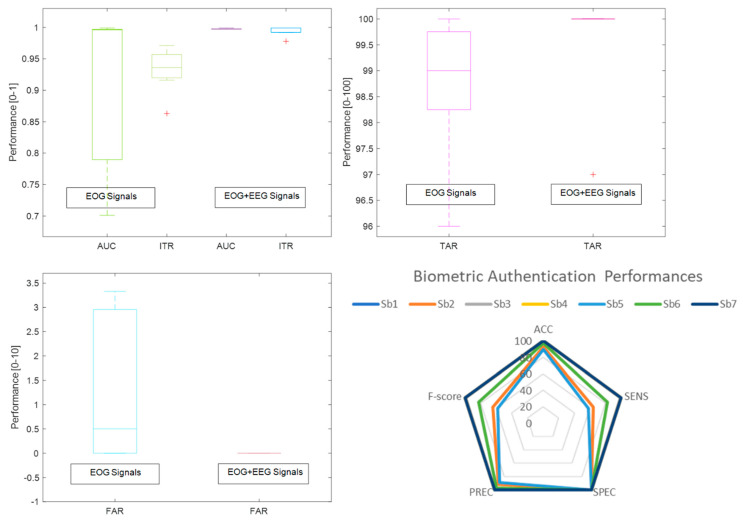
Variations of biometric authentication of AUC/ITR outcomes and TAR/FAR similarity scores for isolated EOG responses and raw EOGs with EEGs (*Whisker box plot*), Discrimination metrics for isolated EOG responses (*Spider plot*).

**Figure 14 biomimetics-08-00378-f014:**
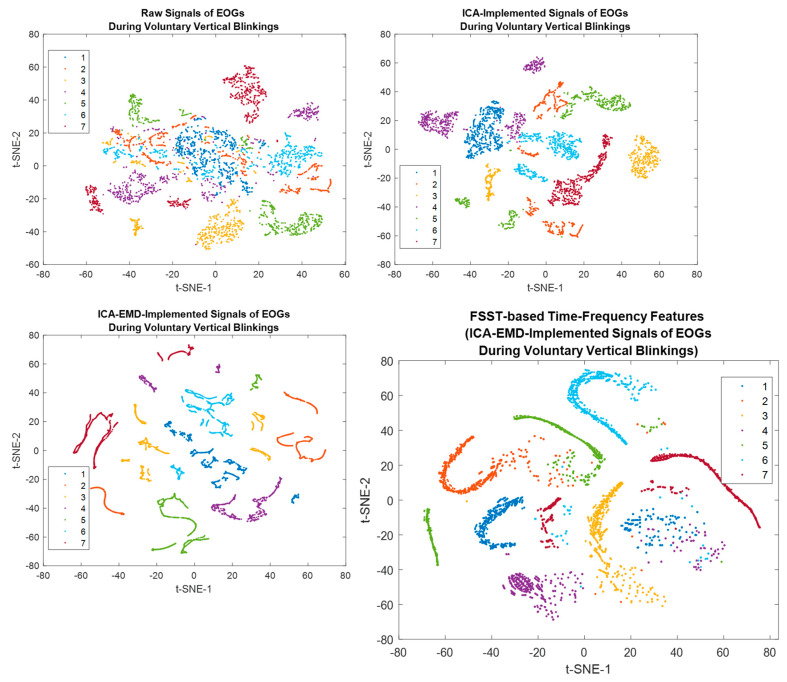
t-distributed stochastic neighbor embedding (t-SNE) plots visualize the separability of the different features (*Each colored clusters describe different individuals*).

**Figure 15 biomimetics-08-00378-f015:**
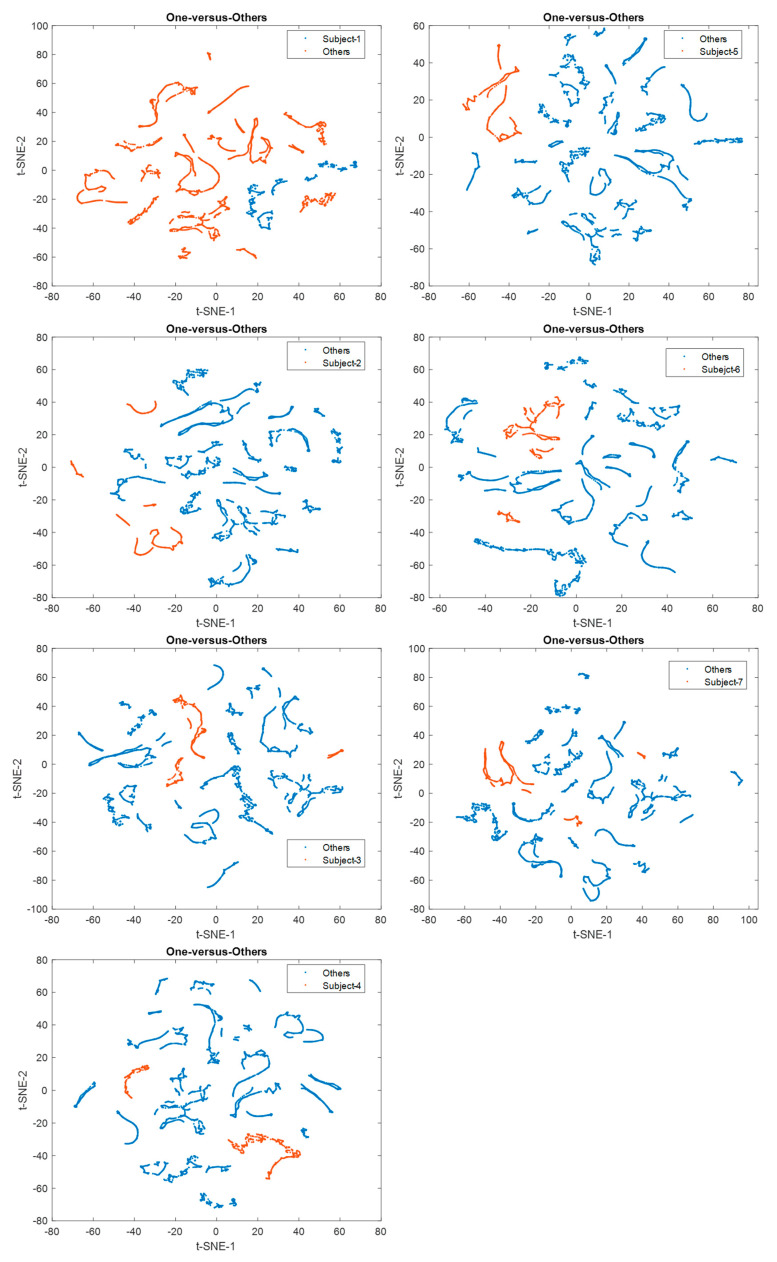
t-SNE plots for the one-versus-others discrimination (ICA-EMD-based datasets were processed for these presentations).

**Figure 16 biomimetics-08-00378-f016:**
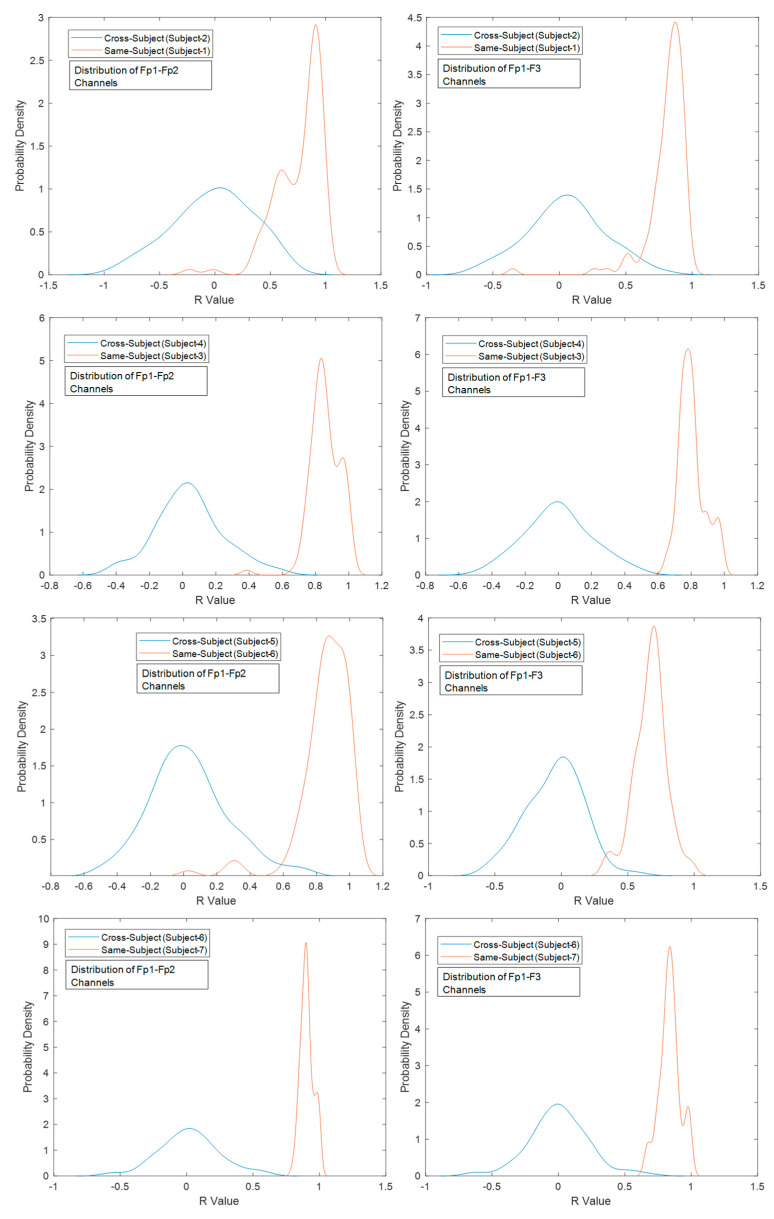
Probability density functions of the correlation coefficients (*R* value) between the frontal-lobe electrodes under cross-subject and same-subject conditions.

**Figure 17 biomimetics-08-00378-f017:**
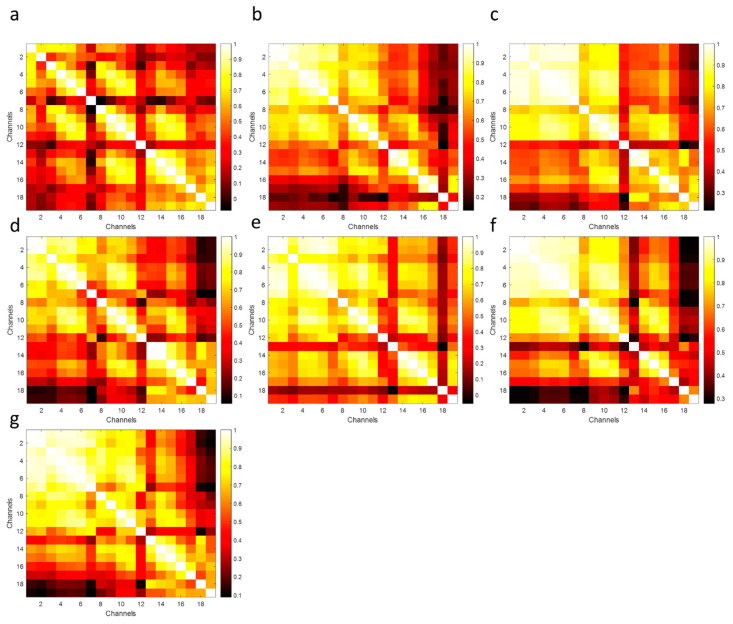
Unique patterns of correlation matrices for each subject related to the electrode locations ((**a**–**g**) *plots correspond to the Subject-1 to Subject-7, respectively*).

**Figure 18 biomimetics-08-00378-f018:**
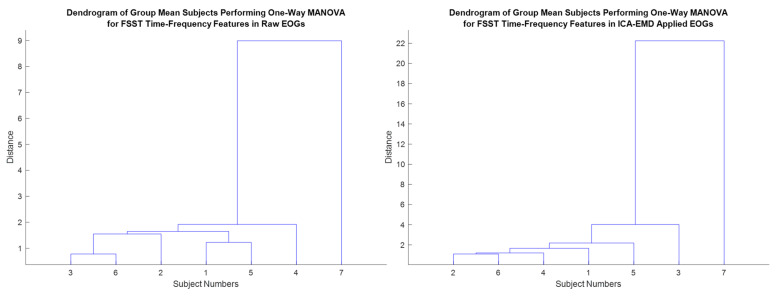
Clustering subjects via MANOVA.

**Figure 19 biomimetics-08-00378-f019:**
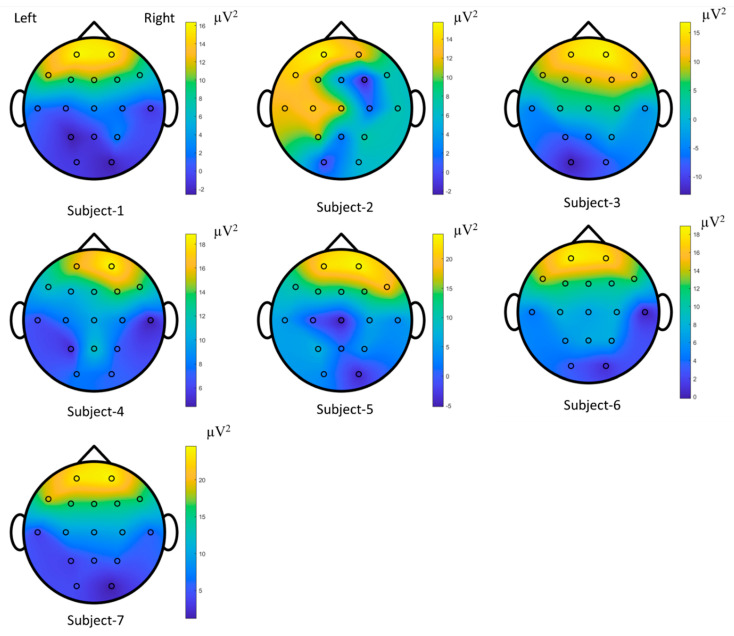
Brain mappings for each individual during the voluntary eye blinkings in EOG-biometric authentication.

**Figure 20 biomimetics-08-00378-f020:**
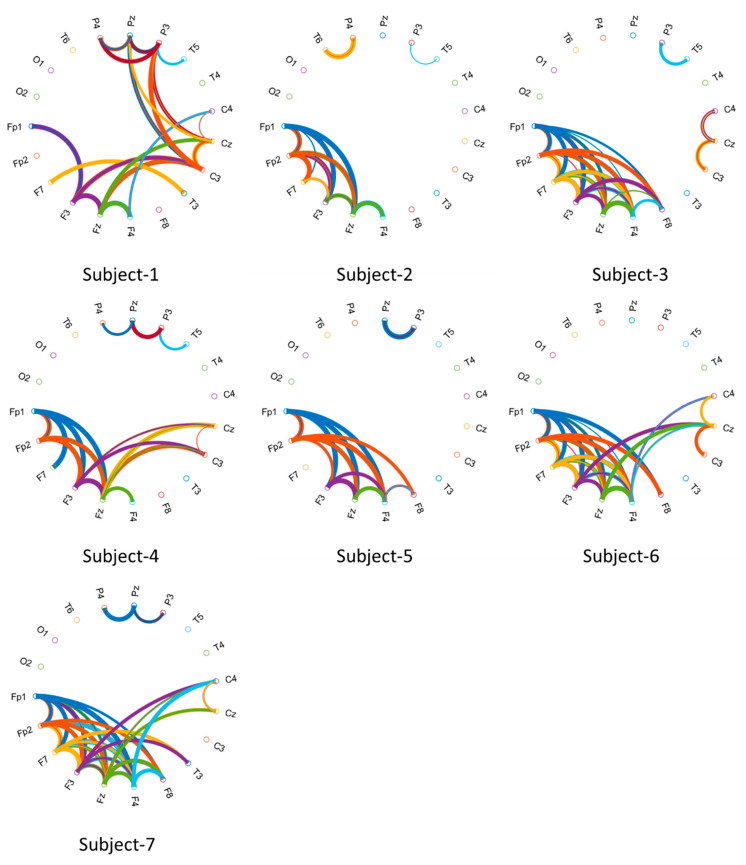
The Circular Graph visualizes the brain’s functional connectivity during the voluntary eye blinkings (Threshold values were applied to draw a clear understanding of functional connectivity over the correlation matrix for each subject).

**Table 1 biomimetics-08-00378-t001:** The list of abbreviations in the paper.

Abbreviations	Descriptions
EOG	Electrooculography
FSST	Fourier Synchrosqueezing Transform
ICA	Independent Component Analysis
EMD	Empirical Mode Decomposition
RNN	Recurrent Neural Network
LSTM	Long Short Term Memory
BiSLTM	Bidirectional LSTM
GRU	Gated Recurrent Unit
EEG	Electroencephalography
EMG	Electromyogram
ECG	Electrocardiogram
TAR	True Acceptance Rate
FAR	False Acceptance Rate

**Table 2 biomimetics-08-00378-t002:** The monopolar placement list for EEG electrodes.

Ch. Num.	1	2	3	4	5	6	7	8	9	10	11	12	13	14	15	16	17	18	19
Ch. Name	Fp1	Fp2	F7	F3	Fz	F4	F8	T3	C3	Cz	C4	T4	T5	P3	Pz	P4	T6	O1	O2

Ch. Num: Channel Number, Ch. Name: Channel Name.

**Table 3 biomimetics-08-00378-t003:** Concise comparative analysis of EOG and EEG-based biometric authentication performances in recent years.

Ref./Year	Signals	Ch. N.	Sb. N.	S./T.	S. M.	Feature Engineering/Models	Performance
[54]/2019	VEP–EEG	9	25	2	Offline	CCA/Task–Related Component Analysis	ACC: 99.43–100%
[55]/2020	VEP–EEG	32	32	32	Offline	Raw, PSD/ CNN–LSTM, CNN–GRU and SVM	ACC: 33.02–100%
[56]/2020	VEP, MI–EEG	19	45	4	Offline	Raw/CNN, CNN–RNN and LDA–SVM	ACC: 77.9–86.9%
[3]/2020	MI–EEG	64	109	10	Offline	EMD, DWT/SVM and Local Outlier Factor (LoF)	TAR: 100% FAR:0.2%
[57]/2021	VEP–EEG	14	70	10	Offline	DWT/CNN–LSTM	ACC: 81.78–91.93% TAR: 92.86% FAR: 29.28%
[35]/2021	VEP–EEG	9	15	1–10	Online/ Offline	Pearson’s Correlation Coefficient/Task–Telated Component Analysis	ACC: 70.27–100%
[58]/2022	MI–EEG	64	109	14	Offline	Gram–Schmidt Orthogonalization Process/CNN	ACC: >98% TAR: >98% FAR: <2%
[40]/2023	MI–EEG	19	7	1	Offline	FSST, ICA, DWT/Ensembled LSTMs	ACC: ≥96.76
[15]/2013	EOG/VOG	≥1	19–40	40	Offline	Linear Discriminant Analysis, Quadratic Discriminant Analysis, Naı¨ve Bayesian Rule, k- Nearest Neighbors	ACC: 53–99%
[14]/2015	EOG	1	25	6–8	Offline	Support Vector Machine, Discriminant Analysis/EMD/Time Delineation Feature Engineering	ACC: ≥97% EER: 3.7%
[13]/2020	EOG	14	46	2	Offline	Gated Recurrent Unit/Time Delineation Feature Engineering	ACC: 98.7%
**Our Study**	EOG	19	7	1	Offline	FSST + ICA + EMD/Ensembled RNNs	ACC: ≥99% TAR: 99.57% FAR: ≤3.33%

**Ch.N:** Channel Number, **Sb.N:** Subject Number, **S.T:** Session/Trial, **S.M:** System Mode, **VEP:** Visual Evoked Potential-EEG, **MI:** Motor Imagery-EEG, **VOG:** Video-Oculography, **EER:** Equal Error Rate.

**Table 4 biomimetics-08-00378-t004:** Biometric approach performances in terms of the discrimination metrics over the EOG signals implemented ICA-EMD techniques.

(%)	Sb1	Sb2	Sb3	Sb4	Sb5	Sb6	Sb7	Avg.
**ACC**	99.92	92.01	99.91	99.93	89.49	97.00	99.99	96.89
**SENS**	99.50	64.13	99.42	99.51	57.65	82.88	99.95	86.15
**SPEC**	100	100	99.99	100	99.92	99.92	100	99.97
**PREC**	99.92	92.01	99.91	99.93	89.49	97.00	99.99	96.89
**F-score**	99.50	64.13	99.42	99.51	57.65	82.88	99.95	86.15
**AUC**	0.997	0.755	0.996	0.997	0.701	0.893	0.999	0.905
**ITR**	0.971	0.916	0.963	0.936	0.938	0.863	0.931	0.931

**Table 5 biomimetics-08-00378-t005:** Biometric approach performances in terms of the matching scores/attempts over the EOG signals implemented ICA-EMD techniques.

		FAR							
(%) *	TAR	Sb1	Sb2	Sb3	Sb4	Sb5	Sb6	Sb7	Avg.
**Sb1**	98	X	0	0	0	0	0	0	0
**Sb2**	99	0	X	1	0	19	0	0	3.33
**Sb3**	100	0	9	X	0	0	2	0	1.83
**Sb4**	100	0	0	0	X	0	0	0	0
**Sb5**	99	0	20	0	0	X	0	0	3.33
**Sb6**	99	0	0	0	3	0	X	0	0.5
**Sb7**	96	0	0	0	0	0	0	X	0
**Avg.**	98.71								

* 100 attempts.

**Table 6 biomimetics-08-00378-t006:** Biometric approach performances in terms of the discrimination metrics over the EOG + EEG signals implemented ICA technique.

(%)	Sb1	Sb2	Sb3	Sb4	Sb5	Sb6	Sb7	Avg.
**ACC**	99.93	99.88	99.93	99.93	99.92	99.91	99.94	99.92
**SENS**	99.53	99.91	99.79	99.54	99.58	99.54	1	99.70
**SPEC**	1	99.87	99.95	99.99	99.98	99.97	99.93	99.96
**PREC**	99.93	99.88	99.93	99.93	99.92	99.91	99.94	99.92
**F-score**	99.53	99.91	99.79	99.54	99.58	99.54	1	99.70
**AUC**	0.997	0.998	0.998	0.997	0.997	0.997	0.999	0.998
**ITR**	0.999	0.992	0.999	0.978	0.993	0.999	0.999	0.994

**Table 7 biomimetics-08-00378-t007:** Biometric approach performances in terms of the matching scores/attempts over the EOG + EEG signals implemented ICA technique.

		FAR							
(%) *	TAR	Sb1	Sb2	Sb3	Sb4	Sb5	Sb6	Sb7	Avg.
**Sb1**	100	X	0	0	0	0	0	0	0
**Sb2**	100	0	X	0	0	0	0	0	0
**Sb3**	100	0	0	X	0	0	0	0	0
**Sb4**	100	0	0	0	X	0	0	0	0
**Sb5**	100	0	0	0	0	X	0	0	0
**Sb6**	97	0	0	0	0	0	X	0	0
**Sb7**	100	0	0	0	0	0	0	X	0
**Avg.**	99.57								

* 100 attemtps.

## Data Availability

The materials used in this research as high-quality EEG files of volunteers can be sent upon the request.

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
