# Peer review of "Fourier Synchrosqueezing Transform-ICA-EMD Framework Based EOG-Biometric Sustainable and Continuous Authentication via Voluntary Eye Blinking Activities"

_biomimetics, 2023, doi:10.3390/biomimetics8040378_

Round 1

Reviewer 1 Report

The manuscript describes a very important topic regarding e-learning. Please specify whether the EEA method applies only to adults. If people over 20 years old have been tested, will this method be suitable for school children?

The author should slightly edit the work. Detailed comments are included in the attached file.

Author Response

All the suggested revisions by Reviewer-1 were made by the author in a careful manner. Thank you for his/her time and effort in improving the article.

Reviewer 2 Report

carefully address the comments 

Need to improve 

Author Response

All the suggested revisions by Reviewer-2 were made by the author in a careful manner. Thank you for his/her time and effort in improving the article.

Round 2

Reviewer 2 Report

carefully address given comments